# Influence of energy gap between charge-transfer and locally excited states on organic long persistence luminescence

Zesen Lin[1,2,3], Ryota Kabe [1,2,3]*, Kai Wang [1,4] & Chihaya Adachi [1,3,5]*

Organic long-persistent luminescence (LPL) is an organic luminescence system that slowly releases stored exciton energy as light. Organic LPL materials have several advantages over inorganic LPL materials in terms of functionality, flexibility, transparency, and solution-processability. However, the molecular selection strategies for the organic LPL system still remain unclear. Here we report that the energy gap between the lowest localized triplet excited state and the lowest singlet charge-transfer excited state in the exciplex system significantly controls the LPL performance. Changes in the LPL duration and spectra properties are systematically investigated for three donor materials having a different energy gap. When the energy level of the lowest localized triplet excited state is much lower than that of the charge-transfer excited state, the system exhibits a short LPL duration and clear two distinct emission features originating from exciplex fluorescence and donor phosphorescence.

[1] Center for Organic Photonics and Electronics Research (OPERA), Kyushu University, 744 Motooka, Nishi-ku, Fukuoka 819-0395, Japan. [2] Organic Optoelectronics Unit, Okinawa Institute of Science and Technology Graduate University, 1919-1 Tancha, Onna-son, Okinawa 904-0495, Japan. [3] JST, ERATO Adachi Molecular Exciton Engineering Project, Kyushu University, 744 Motooka, Nishi-ku, Fukuoka 819-0395, Japan. [4] Institute of Functional Nano and Soft Materials (FUNSOM) and Jiangsu Key Laboratory for Carbon-Based Functional Materials and Devices, Soochow University, Suzhou, Jiangsu 215123, China. [5] International Institute for Carbon Neutral Energy Research (WPI-I2CNER), Kyushu University, 744 Motooka, Nishi-ku, Fukuoka 819-0395, Japan. *email: ryota.kabe@oist.jp; adachi@cstf.kyushu-u.ac.jp

Long-persistent luminescence (LPL), also known as the glow-in-the-dark effect or afterglow, is a phenomenon by which a material emits light for a very long time after the cutoff of photoexcitation[1,2]. The first LPL emitters were based on inorganic crystals and performance was greatly improved through doping[1–3]. Several charge accumulation mechanisms, such as electron or hole trapping mechanisms, have been proposed to explain inorganic LPL[1,3]. Unlike phosphorescence, which can also be long-lived but is a transition between different spin states (usually from a triplet excited state to the singlet ground state), LPL systems do not follow an exponential decay and usually follow a power-law decay because of the presence of the intermediate states (Fig. 1).

Since the mid-1990s, the blending of inorganic LPL materials with a polymer matrix has been the main route for achieving it in commercial applications such as watch dials, fire safety signs, and glow-in-the-dark toys[1–3]. However, inorganic LPL materials exhibit poor compatibility and transparency in common polymers. Moreover, the most efficient inorganic LPL materials nearly all contain rare earth elements such as Sr, Eu, and Dy[1,2,4].

Recently, we realized the LPL emission from purely organic-based materials, including organic small molecules and polymers[5,6]. These organic LPL (OLPL) materials can be easily fabricated by mixing an electron donor and an electron acceptor using various methods such as melt-casting, spin coating, or thermal evaporation[7]. Moreover, the emission color of OLPL systems can be controlled by the addition of dopants[8]. However, a large performance gap still exists between the present OLPL system and the commercial high-performance inorganic LPL products (Supplementary Fig. 1).

The OLPL emission originates from the charge-transfer (CT) transition of a photo-generated exciplex formed between a donor and an acceptor. Some exciplexes can dissociate to form partially charge-separated (CS) states with very long times. The slow recombination of these separated charge carriers leads to continuous emission for over 1 h at room temperature. The LPL process is governed by the recombination of dissociated radical cations and anions with a power-law emission decay[5,9,10], so that the emission duration of OLPL materials is significantly longer than that of conventional room-temperature phosphorescence, which is ideally a first-order reaction with an exponential emission decay[1,2,11,12]. The power-law kinetic results (power-law kinetic, $I(t) \sim t^{-m}$, $m = 0.1–2$) from charge recombination can be explained by several physical models discussed in previous

literatures about LPL from organic molecules ($N,N,N',N'$-tetra-methylbenzidine (TMB)/poly(alkyl methacrylate)s)[13] and thermoluminescence of the inorganic LiF[14], and the organic molecule polyethylene terephthalate[15]. These models can be separated into the diffusion model[9,10,16,17] and the electron tunneling model[18] of geminate ion recombination. In the diffusion model, we consider the distribution of electrons (radical anions) after the charge separation process. The electron tunneling model is mainly used to explain the isothermal recombination luminescence at low temperatures for irradiated organic compounds.

Although a molecule's chemical structure greatly influences its optical and mechanical properties such as absorption and emission spectra, flexibility, and biocompatibility, the strategy for the design and selection of molecules for the OLPL system still remains unclear. We have noted in a previous report that the LPL process, which proceeds through charge dissociation and subsequent recombination, might be affected by the excited-state energy levels of the donor and acceptor, and the exciplex formed between them[6]. However, more detailed relationships are still needed to unlock ways to improve the performance of OLPL materials.

Herein, we demonstrate that the energy gap between the lowest singlet excited state of the exciplex ($^1$CT) and the lowest triplet excited state of the donor ($^3$LE$_D$) strongly affects OLPL performance. Changes in the OLPL properties and the emission mechanism are systematically investigated for three donor materials having similar molecular structures but different energy levels. Optimization of excited-state energy levels based on the uncovered relationships between energy levels and performance will aid the development of efficient OLPL systems aiming for future applications.

## Results

**OLPL materials**. The OLPL systems were fabricated by the melt-casting of a mixture containing 1% of an electron donor and 99% of an electron acceptor[7]. The electron donors, TMB, $N,N'$-dimethyl-$N,N'$-ditolylbenzidine (DMDTB), and $N,N,N',N'$-tetra-tolylbenzidine (TTB), and the electron acceptor, 2,8-bis(diphenylphosphoryl)dibenzo[b,d]thiophene (PPT), are shown in Fig. 2. The highest occupied molecular orbital (HOMO) levels of the donors were determined from the first redox peaks of cyclic voltammograms (Supplementary Fig. 2) and the HOMO levels of

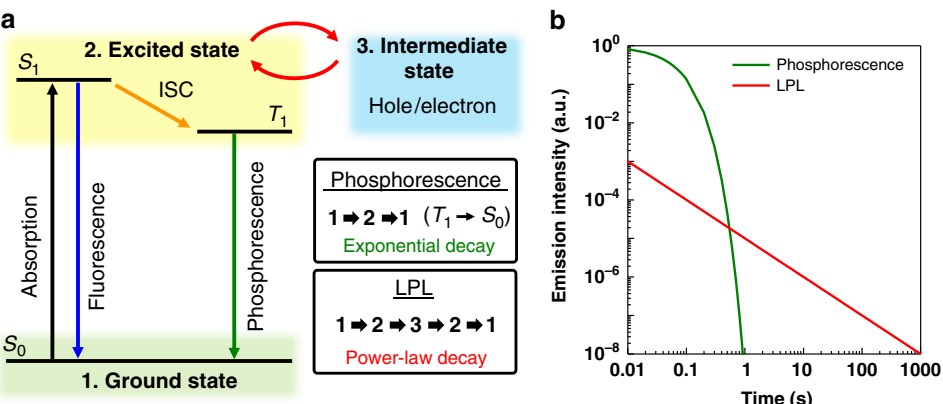

**Fig. 1 Differences between LPL and phosphorescence. a** Schematic diagram of fluorescence, phosphorescence, and LPL. Phosphorescence is a transition from the triplet excited state (T$_1$) to the singlet ground state (S$_0$). LPL is an emission mechanism in which the energy passes through an intermediate state such as a trapped state. There is no restriction regarding the spin state. Although LPL is long-lived because of charge separation and subsequent slow recombination (second-order kinetics) of initially generated excitons, phosphorescence is long-lived because of the low probability of the transition (first-order kinetics) occurring in the initially generated excitons. **b** The ideal emission decay profiles of phosphorescence and LPL on logarithmic plots. Phosphorescence follows an exponential decay and LPL a power-law decay.

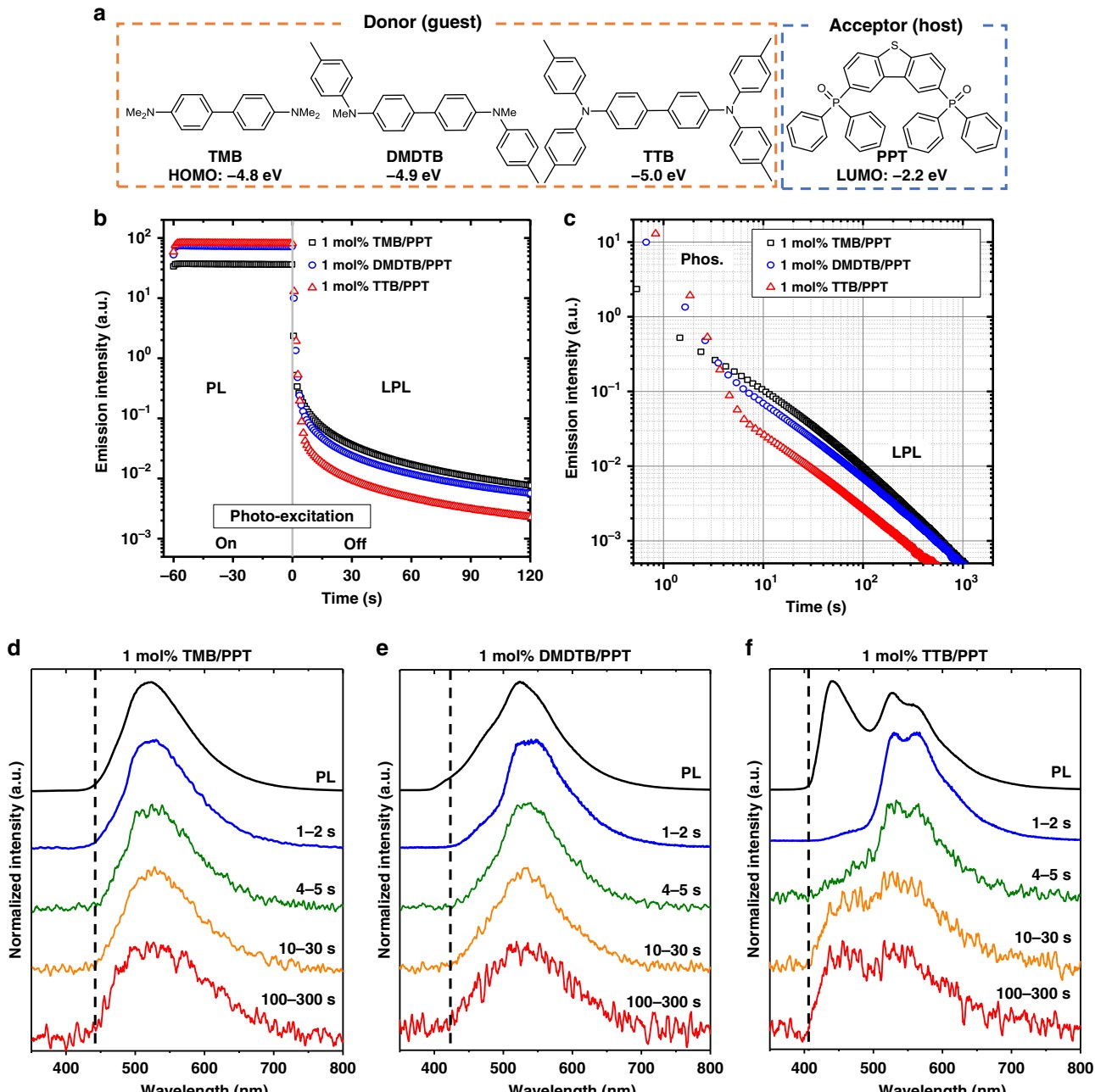

**Fig. 2 Photoluminescence and LPL characteristics of the OLPL systems. a** Chemical structures and HOMO or LUMO energy levels of the three electron donors (TMB, DMDTB, and TTB) and the electron acceptor (PPT). **b**, **c** Semi-logarithmic plots (**b**) and logarithmic plots (**c**) of the emission decay profiles of TMB/PPT, DMTDB/PPT, and TTB/PPT at 300 K. Emission decay profiles contain all of the emission over wavelengths from 400 to 900 nm. Samples were excited for 60 s (from −60 to 0 s) by a 340 nm LED source. PL means the steady-state photoluminescence, LPL means the long-persistent luminescence, and Phos. means the phosphorescence. **d**–**f** The steady-state photoluminescence and time-resolved photoluminescence spectra of 1 mol% TMB/PPT (**d**), DMDTB/PPT (**e**), and TTB/PPT (**f**) films at 300 K. The time-resolved spectra were integrated over periods of 1-2, 4-5, 10-30, and 100-300 s after stopping excitation. The dashed lines indicate the onset of the LPL spectra.

DMDTB (−4.9 eV) and TTB (−5.0 eV) were found to be slightly deeper than that of TMB (−4.8 eV) because of the π-extension provided by the substituent benzene rings.

**Photoluminescence and LPL performances**. The LPL performance of these donor/acceptor systems greatly depends on the donor. Figure 2 shows the steady-state photoluminescence and time-resolved (1–2 s, 4–5 s, 10–30 s, and 100–300 s after stopping excitation) emission spectra of these LPL systems. TMB/PPT and

DMDTB/PPT systems showed a slight change of spectral width with the passage of time. On the other hand, the TTB/PPT system exhibits apparent spectral transformation, i.e., two emission peaks, within 10 s after excitation cutoff, indicating the presence of a second emission process. The emission decay profiles of all of the systems are inverse-power functions of time $t^{-m}$ ($m$ = 0.9–1.3) after 10 s (Fig. 2c and Supplementary Table 1). This non-exponential decay behavior indicates that the LPL emission originates from intermediate CS states[19]. The presence of CS states was proved by using transient absorption measurements in our

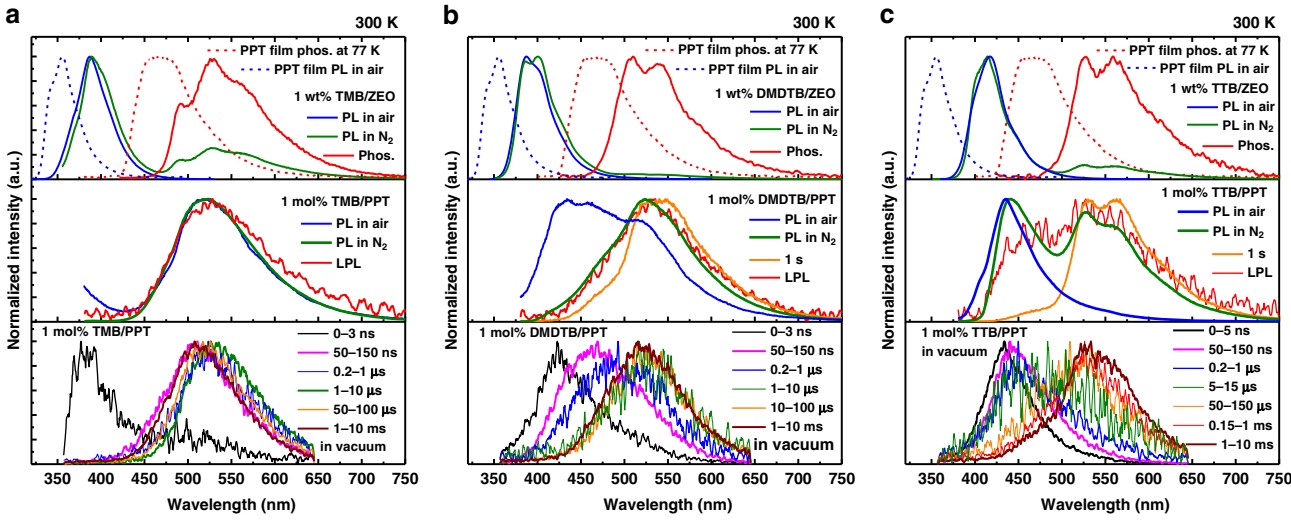

**Fig. 3 Study-state and time-resolved emission spectra of the OLPL systems. a** Fluorescence and phosphorescence spectra (top), steady-state photoluminescence (PL), and LPL spectra (middle), and time-resolved emission spectra (bottom) of TMB/PPT. **b** DMDTB/PPT. **c** TTB/PPT. Optical properties of TMB, DMDTB, and TTB were obtained in ZEONOR films at 300 K. Fluorescence and phosphorescence spectra of PPT were obtained from a neat thin film at 300 K and 77 K, respectively. Time-resolved emission spectra were obtained from streak images at 300 K.

**Table 1 Photophysical properties of the donors, acceptors, and OLPL systems.**

| Sample | HOMO [eV][a] | LUMO [eV][a] | $\Phi_{PL}$ | $\tau_{flu.}$ [NS] | $\tau_{phos}$ [s] | $^1LE_D$ or $^1LE_A$ [eV][b] | $^3LE_D$ or $^3LE_A$ [eV][b] | Sample | $\Phi_{PL}$ | $^1CT$ [eV][c] | $\Delta E(^1CT - ^3LE_D)$ [eV] |
|---|---|---|---|---|---|---|---|---|---|---|---|
| TMB | −4.8 | | 52%[d] | 9.92[d] | 1.43[d] | 3.56[d] | 2.63[d] | TMB/PPT | 24% | 2.79 | 0.16 |
| DMDTB | −4.9 | | 50%[d] | 1.76[d] | 0.79[d] | 3.43[d] | 2.62[d] | DMDTB/PPT | 27% | 2.87 | 0.25 |
| TTB | −5.0 | | 47%[d] | 1.30[d] | 0.72[d] | 3.25[d] | 2.50[d] | TTB/PPT | 28% | 3.04 | 0.54 |
| PPT | | −2.2 | 1%[e] | 1.15[e] | 1.01[f] | 3.76[e] | 2.91[f] | | | | |

[a]Calculated from CV or DPV peaks
[b]Calculated from the onset of the emission spectra
[c]Calculated from the onset of the LPL spectra
[d]In ZEONOR film at 300 K
[e]In the neat film at 300 K
[f]In the neat film at 77 K

previous publication[5] and time-resolved electron spin resonance (ESR) measurements. The ESR signal attributed to the organic radicals clearly increases after photoexcitation and gradually decreases by time (Supplementary Fig. 3).

To understand the detailed emission mechanisms, we obtained time-resolved emission spectra from the OLPL systems on nanosecond through millisecond timescales by using a streak camera (Fig. 3 and Supplementary Fig. 4–9). Also, the fluorescence and phosphorescence spectra of the donors were obtained from 1 wt% donor-doped films of the cyclic olefin copolymer ZEONOR, which acts as a nonpolar solid matrix that does not form a CT complex with the donor dopants[20]. The optical properties of the donors doped in ZEONOR films are almost identical to those in toluene solutions, indicating that there are no aggregation or polarization effects (Supplementary Fig. 10). Moreover, as the solid-state matrix can prevent the nonradiative deactivation of the dopants, the room-temperature phosphorescence of the donors can be obtained. The phosphorescence decays of the donors (Supplementary Fig. 10) are exponential with lifetimes of 1.43 s (TMB), 0.79 s (DMDTB), and 0.72 s (TTB). The fluorescence spectra were obtained in air, which quenches the photo-generated triplet excitons because of the presence of oxygen. Optical properties and the energy levels calculated from the onsets of the emission spectra are summarized in Table 1.

The time-resolved emission spectra of TMB/PPT system indicate the presence of weak fluorescence from TMB for at least 3 ns after excitation cutoff (Fig. 3a and Supplementary Fig. 5). This fluorescence originates from the TMB molecules, which do not form CT with PPT. After the initial fluorescence of TMB, an exciplex emission that slightly shifts with time was obtained. The temperature dependencies of the time-resolved emission spectra and the emission decay profiles on a microsecond timescale clearly indicate the presence of thermally activated delayed fluorescence (TADF), which is often obtained from exciplex systems (Supplementary Fig. 7)[21,22]. The spectral shift of the exciplex emission can be explained by the large dipole moment of PPT. As the CT excited states have large dipole moments (Supplementary Table 2), reorganization of the PPT matrix in the excited-state— so-called solid-state solvation—induces the spectral shift during the TADF process[23–25].

**Emission mechanism**. The proposed emission mechanism and energy diagrams obtained from the onsets of the emission spectra are shown in Fig. 4. As the lowest triplet excited-state emission of the exciplex ($^3CT$) could not be obtained directly, we assume that $^3CT$ is almost identical to the lowest singlet excited-state of the exciplex ($^1CT$), because excellent separation of the HOMO and the LUMO orbitals on the donor and acceptor, respectively, of the

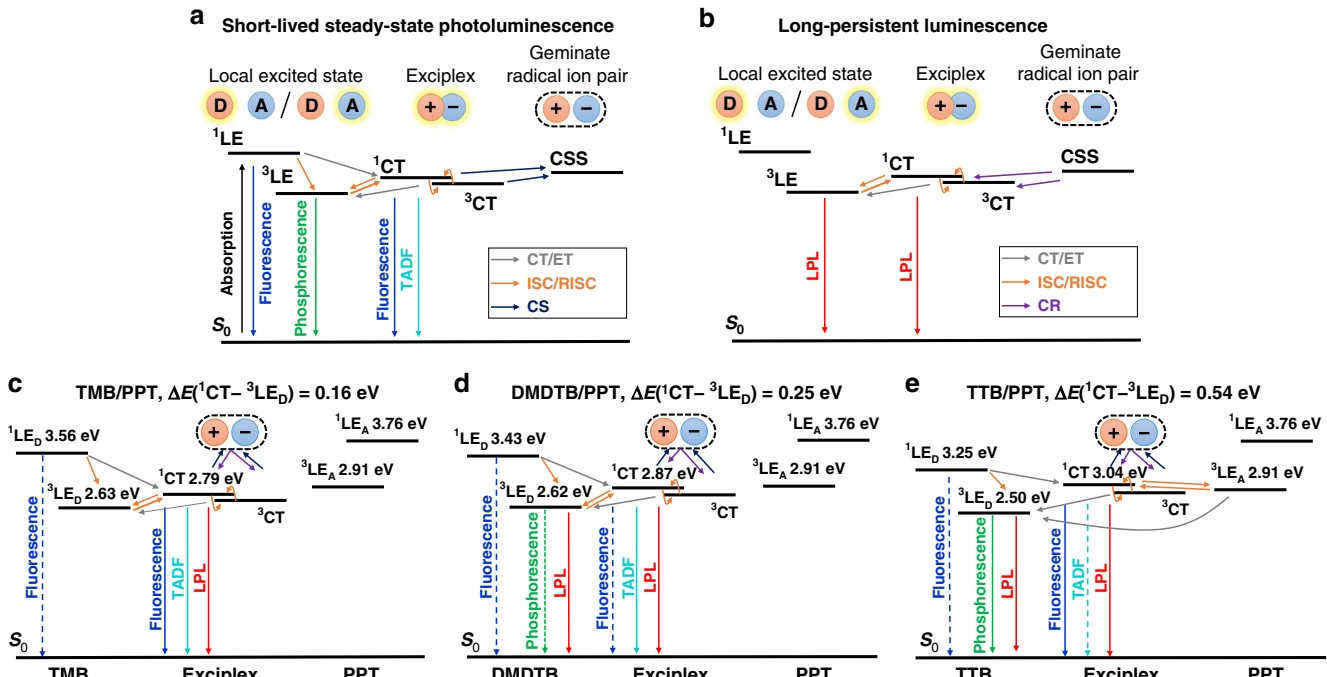

**Fig. 4 Proposed emission mechanism of the OLPL systems.** Proposed emission mechanism before (**a**) and after (**b**) recombination of charges and LPL path in TMB/PPT (**c**), DMDTB/PPT (**d**), and TTB/PPT (**e**). The energy levels were calculated from the onsets of the corresponding emission spectra. Abbreviations of electron donor (D), acceptor (A), charge transfer (CT), electron transfer (ET), charge-separated state (CSS), charge separation (CS), and charge recombination (CR) are used. The dotted lines represent weaker luminescence processes and the solid lines stronger ones.

exciplex induces a small energy gap between $^1$CT and $^3$CT[21,22,26,27]. Recent studies of TADF molecules indicate that the locally excited triplet state of donor or acceptor units, which are analogous to the triplet states located on the donor or acceptor molecule in an exciplex, contribute the reverse intersystem crossing (RISC) process; thus, this is one important factor we considered when investigating the emission mechanism[28–31].

**OLPL from TMB/PPT system.** In the case of TMB, the lowest triplet excited-state of the donor ($^3$LE$_D$) is slightly lower than the lowest singlet CT excited-state of the exciplex ($^1$CT). This relatively small energy gap between $^3$LE$_D$ and $^1$CT ($\Delta E(^1$CT − $^3$LE$_D) = 0.16$ eV) enhances the TADF activity through the processes of ISC and RISC. Although the emission decay of an ideal TADF material follows a bi-exponential decay consisting of a fast fluorescence component and a slow TADF component, the emission decay of TMB/PPT is mostly dominated by the power-law decay originating from the charge recombination process and, therefore, does not exhibit bi-exponential decay. The emission decay profiles and the corresponding time-resolved spectra indicate that the TMB/PPT system exhibits fluorescence from TMB, prompt fluorescence from exciplexes, TADF from exciplexes, and finally LPL emission from exciplexes via charge separation and recombination processes, successively.

**OLPL from TTB/PPT system.** The steady-state photoluminescence spectrum of the TTB/PPT system exhibits clear two distinct emission features with the peaks at 440 nm and 530 nm originating from exciplex fluorescence and donor phosphorescence, respectively (Fig. 2f). Although the emission peak at 440 nm is close to the peaks of PPT phosphorescence and TTB fluorescence, this peak can be attributed to exciplex fluorescence for two reasons. First, PPT phosphorescence is quenched by oxygen, but the TTB/PPT film in air still exhibits a similar peak at 435 nm. Second, the emission

decay at 440 nm of TTB/PPT is much longer than that of TTB fluorescence. The sharpness of the exciplex emission can be ascribed to the self-absorption by the radical cation species of TTB (Supplementary Fig. 11). The contribution of delayed fluorescence by triplet–triplet annihilation is almost negligible, as the donor concentration is only 1% and the phosphorescence timescale is much shorter than that of LPL. The TTB concentration dependence of the emission spectra and emission decay profiles (Supplementary Fig. 12) are also consistent with exciplex emission. The LPL duration becomes shorter at higher concentrations of the donor, because the accumulated changes can more easily recombine with donor molecules. For higher donor concentrations, the exciplex emission was slightly redshifted and the room-temperature phosphorescence from donors became weaker because of aggregation of donor molecules.

In the case of the TTB/PPT system, the TADF process is almost completely quenched, because the $^3$LE$_D$ of TTB is much lower than the $^1$CT. This large energy gap ($\Delta E(^1$CT − $^3$LE$_D) = 0.54$ eV) suppresses the RISC process, so that photo-generated excitons are becomes trapped on $^3$LE$_D$, leading to room-temperature phosphorescence from TTB. As the energy gap between the $^1$CT and the lowest triplet excited-state of the acceptor ($^3$LE$_A$) is small enough, these energy levels should contribute to the TADF process. However, the generated $^3$LE$_A$ excitons can easily decay to the lower $^3$LE$_D$[32,33]. As the TADF process is suppressed at low temperatures, the emission spectra at 10 K contain stronger phosphorescence components from TTB than the spectrum at room temperature (Supplementary Fig. 13). Further, after the decay of phosphorescence from the TTB triplet excited states, the emission occurs from the excitons generated by the CS state. As charge recombination generates both singlet ($^1$CT) and triplet ($^3$CT) exciplexes, the LPL emission consists of both exciplex fluorescence from $^1$CT and donor phosphorescence from $^3$LE$_D$, which is populated by the transfer of excitons from $^3$CT to $^3$LE$_D$. Because of the dual emission from $^1$CT and $^3$LE$_D$, TTB/PPT

system exhibits white emission. The CIE coordinates ($CIE_x$, $CIE_y$) of the steady-state photoluminescence and LPL spectra are (0.27, 0.33) and (0.31, 0.37), respectively (Supplementary Fig. 14 and Supplementary Movie 1).

**OLPL from DMDTB/PPT system**. The energy gap $\Delta E(^1CT - ^3LE_D) = 0.25$ eV of the DMDTB/PPT system is between those of TMB/PPT and TTB/PPT systems. Therefore, DMDTB/PPT system also exhibits dual emission from both exciplex fluorescence ($^1CT$) and DMDTB phosphorescence ($^3LE_D$). The exciplex emission of DMDTB/PPT shows a large spectral shift during the TADF process. This large emission shift would be derived from the excited-state conformational change between the structural conformers of DMDTB (Supplementary Fig. 15)[24]. Because of this large spectral shift, the exciplex fluorescence and DMDTB phosphorescence have a large spectral overlap. The lack of spectral shift and TADF emission in the LPL emission spectrum at 10 K confirms the contribution of TADF to the emission of the DMDTB/PPT system at room temperature.

## Discussion

These results clearly indicate the importance of the energy level of $^3LE_D$ for obtaining efficient LPL emission. As $^3LE$ excitons are less likely to undergo the charge-transfer step needed for creating separated charges that contribute to LPL, the higher exciton population on $^3LE_D$ induced by a large energy gap of $\Delta E(^1CT - ^3LE_D)$ will reduce the number of excitons that can convert into CS states. Thus, efficient LPL emission requires a small energy gap to ensure a higher number of $^1CT$ excitons that can contribute to the accumulation of separated charges.

Notably, the presented photoluminescence quantum yields ($\Phi_{PL}$) do not completely reflect the LPL components (Table 1). The quantum efficiency of LPL emission is difficult to define, because the charge accumulation and release processes are slow and complicated in contrast to those of long-lived phosphorescence. Furthermore, the LPL emission depends on the excitation time as well as the excitation power, whereas the phosphorescence component is constant (Supplementary Fig. 16 and Supplementary Fig. 17). As the LPL system continuously provides the new excited states after turning off the photoexcitation, we cannot calculate the $\Phi_{PL}$ from the steady-state photoluminescence spectra (Supplementary Fig. 18). This is why $\Phi_{PL}$ is not discussed even in inorganic LPL materials[1–3].

In conclusion, we demonstrated that the $^3LE_D$ influences LPL emission by changing the energy gap of $\Delta E(^1CT - ^3LE_D)$. When the energy level of $^3LE_D$ is significantly lower than that of the $^1CT$, the OLPL efficiency was reduced. As a large energy gap induces a higher $^3LE_D$ population through ISC and energy transfer from $^3CT$ to $^3LE_D$, the emission from both $^1CT$ and $^3LE_D$ contributed to LPL. This dual emission from both $^1CT$ and $^3LE_D$ produced white light without the use of additional dopants. Moreover, we found that absorption by radical cation species generated by the charge separation process also affects the LPL emission spectra. Future efficient OLPL systems using both small molecules and polymers will be developed based on these considerations.

## Methods

**Materials**. TMB and TTB were purchased from TCI Chemicals (Tokyo, Japan). DMDTB was synthesized according to Supplementary Methods. PPT was prepared as described in the literature. All materials were purified by recrystallization and sublimation, and were stored in amber bottles in a glovebox. ZEONOR 1060R was obtained from ZEON Japan (Tokyo, Japan). Other materials were used as received. Inorganic LPL product was obtained from LTI Corporation (Kyoto, Japan).

**Film fabrication**. Thick films (0.4 mm) for the optical measurement were fabricated by a melt-casting method[7]. Mixed materials were heated up the melting point of the acceptor (250 °C) in a nitrogen-filled glovebox. After melting, the substrate was cooled rapidly to room temperature. Thin films for the UV-visible absorption measurements were fabricated by sandwiching the heat-melted materials between two quartz substrates. Film thickness were $18 \pm 4\ \mu m$ (PPT), $25 \pm 3\ \mu m$ (TMB/ PPT), $7 \pm 4\ \mu m$ (DMDTB/PPT), and $16 \pm 8\ \mu m$ (TTB/PPT). The ZEONOR doped films were fabricated by solution processing[6]. Materials were dissolved in xylene by ultrasonication and drop-cast on the substrate at 80 °C and then annealed for 1 h at 170 °C in a nitrogen-filled glovebox.

**Characterization**. $^1H$ nuclear magneticresonance (NMR) (Supplementary Fig. 19) and $^{13}C$ NMR spectra (Supplementary Fig. 20) were recorded with a Bruker AVANCE III 500 MHz spectrometer. Molecular weight was measured in positive-ion atmospheric-pressure chemical ionization mode on a Waters 3100 mass detector (APCI-MS). Elemental analysis (C, H, and N) was carried out with a Yanaco MT-5 elemental analyzer. Film thicknesses were measured in five different positions on each film using a micrometer screw gauge and averaged. The cyclic voltammetry (CV) and differential pulse voltammetry (DPV) measurements were carried out using an electrochemical analyzer (Model 608D + DPV, BAS). The measurements were performed in dried and oxygen-free $CH_2Cl_2$ using 0.1 M tetrabutylammonium hexafluorophosphate ($TBAPF_6$) as a supporting electrolyte. A platinum fiber was used as a working electrode, glassy carbon as a counter electrode, and $Ag/Ag^+$ as a reference electrode. Redox potentials were referenced against ferrocene/ferrocenium ($Fc/Fc^+$). The CV curves were recorded at a scan rate of 100 mV s$^{-1}$ and the DPV curves were obtained with a pulse width ($\Delta E_{pulse}$) of 0.2 s. The HOMO energy levels of the three donors were calculated according to the equations of $E_{HOMO\ or\ LUMO} = -E_{redox}$(vs. Fc/ $Fc^+$) – 4.8 eV[34] and $E_{redox} = E_{peak} + \Delta E_{pulse}/2$, where $E_{redox}$ and $E_{peak}$ are the formal electrode potentials and the DPV peak potentials of the redox, respectively. The LUMO energy level of PPT was calculated from the CV data in DMF[8].

**Optical measurements**. The absorption spectra were recorded on a UV-vis-NIR spectrophotometer (LAMBDA 950, Perkin Elmer). The absorption spectra of the radical species were obtained under electrical oxidation in a solution containing 0.1 M $TBAPF_6$. The photoluminescence spectra in air were recorded on a spectrofluorometer (FP-8600, JASCO). The phosphorescence spectra at 77 K were recorded on a multi-channel spectrometer (PMA-12, Hamamatsu Photonics) excited using a 340 nm LED (M340L4, Thorlabs) with a bandpass filter (340 ± 5 nm). The absolute photoluminescence quantum yields ($\Phi_{PL}$) were measured using a quantum yield spectrometer (C9920-02, Hamamatsu Photonics). The streak images, transient photoluminescence spectra, and decay profiles on various timescales were measured in vacuum using a streak camera system (C4334, Hamamatsu Photonics) equipped with a cryostat (GASESCRT-006-2000, Iwatani) and excitation was provided by a nitrogen gas laser (KEN-X, USHO). LPL performance was obtained using a homemade measurement setup with an excitation power of 230 μW and an excitation duration of 60 s[8]. Supplementary Movie 1 was recorded on a Sony α7sII digital camera with 1 mol % TTB/PPT film excited by 365 nm UV lamp for 5 min at 300 K.

**ESR measurements**. The ESR spectra were recorded on a JEOL JES-FA200 spectrometer. The samples were photo-excited by using a 340 nm LED (M340L4, Thorlabs).

## Data availability

The data that support the findings of this study are available from the corresponding author upon reasonable request.

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

## Acknowledgements

This work was supported by the Japan Science and Technology Agency (JST), ERATO, Adachi Molecular Exciton Engineering Project, under JST ERATO Grant Number JPMJER1305, Japan; the International Institute for Carbon Neutral Energy Research (WPI-I2CNER) sponsored by the Ministry of Education, Culture, Sports, Science, and Technology (MEXT); JSPS KAKENHI Grant Numbers JP18H02049 and JP18H04522; and the Mitsubishi Foundation. Z.L. was supported by the Japanese Government (MEXT) Scholarship and also acknowledges the MEXT Top Global University Project and the China Scholarship Council (CSC). We thank Dr. W. Potscavage for his assistance in writing this manuscript.

## Author contributions

R.K. proposed the project. Z.L. designed molecules and perfomed all experiments. Z.L. and K.W. performed the computational caluculararion. R.K. and C.A. supervised the project. Z. L., R.K., and C.A. wrote the manuscript and all authors reviewed the manuscript.

## Competing interests

The authors declare no competing interests.
