## [Peer Review File · Nature Communications]

Reviewers' comments:

Reviewer #1 (Remarks to the Author):

Comments on NCOMMS-19-12867

Recent years, organic long-persistent luminescent (LPL) and room-temperature phosphorescent materials have been paid much attention. In this work, Adachi and co-workers found that energy gap between the lowest localized triplet excited-state and the lowest singlet charge-transfer excited-state in exciplex system plays an important role in achieving LPL. The detailed mechanism has been studied by both theoretical and experimental work. Thus, these results are very interesting, which may arouse broad readership of Nature Communications in the fields of luminescent materials and organic chemistry. Because of the very high level of Nature Communications, the level of the manuscript can be further enhanced to meet the high quality before publication. The details are shown below:

1. Authors have illustrated that the PLQY is very important for application. So, they need to add more information about PLQY of these LPL systems, particularly for the white-light materials. Also, they can compare the PLQY with some similar systems.
2. Authors can discuss more about how to obtain suitable energy gap for the design of new LPL materials.
3. I noted that the content of donor is 1% in all the work. Whether the different ratios of donor/acceptor influence on the LPL performance?
4. More discussion on the comparison between experimental and computational DFT studies can be provided.
5. To arouse broader interest from the readership in this field, several strong related works can be added, such as donor/acceptor triplet energy transfer (*Adv. Funct. Mater.* 2019, 29, 1807599; *Chem. Sci.* 2017, 8, 590); long-lasting luminescence (*Nat. Commun.* 2018, 9, 2798). These literatures may help readers better understand the development of recent similar luminescent materials.
6. Authors need to discuss the stability of the organic long-persistent luminescent LPL.
7. Several mistakes can be avoided, such as page 12, line 242: Wiley Online Library?

Overall, from scientific and technological views, this work can be suitable for Nature Communications after giving revisions based on the comments above.

Reviewer #2 (Remarks to the Author):

In this work, the authors suggest the modulation of energy between exciplex state and localized triplet state influences the performance of long-persistent luminescence (LPL). Through the molecular design of the three donors with different energy of HOMO in exciplex system, distinct excited-state pathways show the competition between phosphorescence and exciplex emissions. Though the experimental results seem to be self-consistent with the model, the proposed mechanism is not new. The exciplex systems based on PPT and TMB have been reported in several articles and similar mechanism has been discussed in detail (ref: Adv. Mater. 2018, 30, 1800365; Adv. Mater. 2018, 30, 1803713. and Chem. Lett. 2019, 48, 270-273). Furthermore, organic long persistence luminescence has been intensively studied during the past few years. Numerous papers have been published, which all claimed to have a bright future goal for advanced applications such as display, anti-fake, information, storage and bio-imaging etc. Unfortunately, up to this stage, I do not see any breakthrough in the practical or leading applications. Note that various other types of inorganic phosphors possess better efficiency of LPL in similar applications proposed. In other words, a solid progress and task of top priority of the relevant research in this field is to show impressive and leading applications, not the explorations of more molecules and perhaps debatable mechanisms but with similar properties. These, together with a number of technical comments listed below, lead me to hesitate recommending this article published in Nature Communication.

Comments

1. The system of TTB/PPT seems to have no solid evidence for the existence of exciplex. If the emission at 435 nm is ascribed to the exciplex fluorescence, then the authors have to explain the first fast decay (0~5 ns) accompanied with the later slow rise (10~30 s) in time resolved emission spectra. To my viewpoint, it seems to be more appropriate that the emission at 435 nm is from singlet LE of donor rendered perhaps by triplet-triplet annihilation (rather slow diffusion in solid) of triplet LE of donor because emission of TTB/ZEO also exhibits singlet emission at 435 nm. If this is the case, it is irrelevant to the exciplex system.
2. The definition of the normalized intensity for transient emission decay curves in Figure S6 and S7 is vague. In Figure S7c, the intensity of DMDTB/PPT and TTB/PPT at 10 μ s have dropped to the order of 10^{-4} already. However, the order of the intensity in Figure S7f are still 10^{-2} to 10^{-3} at 10 ms, which is inconsistent and contradicts to the results in Figure S3 to S5. The author should explain this confliction.
3. The intensity of the all samples in Figure 1c decays to the order 10^{-2} at 10 to 100 sec. I doubt the normalized intensity ratio can really represent the ratio of emission contributed by LPL since the intensity ratio is different from the transient emission decay profile (Figure S6 and S7). If the intensity ratio in Figure 1c indeed represents the real ratio of LPL, please explain the different results in Figure S6 and S7. If not, the authors have to elaborate how much ratio of emission resulted from LPL.
4. The proposed mechanism shown in Figure 3a is dubious. The rate of reverse intersystem crossing (RISC) has been reported to be about 10^5 to 10^6 s $^{-1}$ in previous literature (Appl. Phys. Lett. 2013,

102, 153306.). From the kinetic point of view, the rate of RISC is much faster than the radiative rate of the LPL (the lifetime is too long). As a result, it is hard to understand why the 3CT state can undergo RISC and emit LPL simultaneously. The RISC should dominate the excited 3CT state behavior. The authors should give a rational explanation.

5. Continuing the above comment, in Figure 3d, the authors claimed the 3LED state would generate both phosphorescence and LPL. The authors should tell the difference between these two emissions. Moreover, the authors carried out the emission spectra in 10 K showing the stronger phosphorescence from TTB species than that at room temperature (Figure S10). This observation indicates that the triplet exciton would be finally trapped in the 3LED state in the TTB/PPT system. Such statement is very tenuous due to the lack of direct evidence. The non-radiative pathways would be suppressed at low temperature, which would certainly enhance the intensities of the phosphorescence from all triplet states. Therefore, this result cannot support the proposed mechanism in a direct manner. More convincing evidence should be provided.

6. Regarding the DMDTB/PPT system, the authors mentioned that the exciplex emission shows a large spectral shift during the TADF process, which can be attributed to the excited-state conformational change between the two stable isomers of DMDTB. It is believed that the cis-trans isomerization is difficult to occur in the solid film. So, what kinds of conformation-dependent emissive properties do the authors expect here? The authors should pay some efforts to make the interpretation clear. Furthermore, the cited reference (ref. 25) is inappropriate because it doesn't provide any related photophysical idea.

Minor

1. Please change the "Figure 2" in page 4 line 80 to Figure 1. "Figure 2 shows the steady-state photoluminescence and time-resolved (1–2 s, 4–5 s, 10–30 s, and 100–300 s after stopping excitation)"

Reviewer #3 (Remarks to the Author):

This manuscript from Adachi and coworkers follows up on this groups very interesting and significant report in 2017 that long-lived luminescence can be obtained from organic exciplexes. The present manuscript examines three combinations of electron donors with an electron acceptor in order to understand how the electronic properties of the donor affect OLPL performance. This is an important study as the results can be used to guide the design of future exciplex systems. The authors reach the conclusion that the energy gap between the lowest localized triplet and lowest charge transfer singlet is an important design criterion, with a larger gap to reduced efficiency. It is not completely clear to this reviewer how this conclusion is supported by the data: In Table 1 the PL

quantum efficiencies of the three donor/acceptor blends are given along with the energy gap. The three blends show rather similar quantum efficiencies, but this is apparently the overall quantum efficiency for PL, not just OLPL. What is the OLPL quantum yield in each of these blends? This is the key parameter assessing performance of the OLPL material. The data shown in Figure 1c suggest that there is a difference in OLPL efficiency, but this is not quantitated, and it is not clear whether these decays are normalized or not. This is a key issue that the authors must address prior to publication.

Other points for the authors to address:

- 1) It would be useful to provide a comparison of the efficiency of these systems relative to the conventional inorganic/polymer systems discussed in the introduction.
- 2) Figure 3 – why are some decays dotted lines and some solid lines? There doesn't seem to be any logic to this and no legend is provided. The Figure caption is also unclear – each frame should be clearly labelled in the caption.
- 3) Lines 178-180 – there are examples of pure organic systems that show RTP – the statement regarding efficiency of these should be referenced.
- 4) Line 80 – Figure 1 not Figure 2.

Response to reviewers' comments

We would like to thank all reviewers for his/her constructive comments to improve this manuscript.

Answers to the reviewer #1

Recent years, organic long-persistent luminescent (LPL) and room-temperature phosphorescent materials have been paid much attention. In this work, Adachi and co-workers found that energy gap between the lowest localized triplet excited-state and the lowest singlet charge-transfer excited-state in exciplex system plays an important role in achieving LPL. The detailed mechanism has been studied by both theoretical and experimental work. Thus, these results are very interesting, which may arouse broad readership of Nature Communications in the fields of luminescent materials and organic chemistry. Because of the very high level of Nature Communications, the level of the manuscript can be further enhanced to meet the high quality before publication. The details are shown below:

Q1. Authors have illustrated that the PLQY is very important for application. So, they need to add more information about PLQY of these LPL systems, particularly for the white-light materials. Also, they can compare the PLQY with some similar systems.

A: We appreciate the comment by the reviewer. Although the photoluminescence quantum yield (PLQY) is an important factor when discussing conventional photoluminescence emitters, defining a similar value, e.g., an LPL quantum yield, is difficult because of the charge accumulation process. This is why quantum efficiencies are not reported for inorganic LPL systems even in academic research.

In the case of the conventional photo emitters, the excited states are generated only by the initial photoirradiation. All emission processes are first-order reactions that follow exponential decay. Therefore, PLQY can be calculated from the area under the absorption and the area under the emission spectra obtained by using an integration sphere. (Figure S17).

Phosphorescence quantum yield can also be calculated from the area under the absorption spectrum and the area under the steady-state emission spectrum, this time only considering phosphorescent emission (Figure S17a). Even if the fluorescent and phosphorescent spectra overlap, we can calculate the ratio of fluorescence and phosphorescence from the emission decay profiles since all processes follow exponential decay (Figure S17b).

In contrast, LPL systems store excited energy through a charge accumulation process and generate new excited states from the accumulated charges over a long period after turning off the photoexcitation. This process is a second-order reaction, so it follows a power-law decay. Since the charge separation and recombination processes are very slow, LPL emission is not fully reflected in the steady-state emission spectra (Figure S17c).

Furthermore, OLPL systems exhibit fluorescence, TADF, phosphorescence, and LPL from similar energy levels. Since the emission decay profile of the OLPL system is very complicated, even in the ideal case, observation of the emission decay profile from nanosecond (10^{-8} s) to tens of hours (10^4 s) timescales are required to calculate the LPL contribution (Figure S17d). On top of that, the LPL component depends on the excitation conditions (time and intensity) (Fig. S15 and S16). Therefore, we cannot provide an LPL quantum efficiency.

We added the following sentences from page 9 line 16.

“Notably, the presented photoluminescence quantum yields (Φ_{PL}) do not completely reflect the LPL components (Table 1). The quantum efficiency of LPL emission is difficult to define because the charge accumulation and release processes are slow and complicated in contrast to those of long-lived phosphorescence. Furthermore, the LPL emission depends on the excitation time as well as the excitation power, while the phosphorescence component is constant (Figure S15 and S16). Because the LPL system continuously provides the new excited states after turning off the photoexcitation, we cannot calculate the Φ_{PL} from the steady-state photoluminescence spectra (Figure S17). This is why Φ_{PL} is not discussed even in inorganic LPL materials.^{1,2,3}”

Figure S17. Ideal emission spectra and logarithmic plots of the emission decay profiles of phosphorescent materials (**a, b**) and OLPL materials (**c, d**). The phosphorescence quantum yield can be calculated from the areas under the absorption and phosphorescence emission spectra (**a**). If the fluorescence and phosphorescence spectra have a large overlap, phosphorescence quantum yield can be estimated from the emission decay profiles because both fluorescence and phosphorescence follow an exponential decay (**b**). The OLPL system exhibits fluorescence, TADF, phosphorescence, and LPL from similar energy levels (**c**). Although the fluorescence, TADF, and phosphorescence follow an exponential decay, the power-law decay of LPL makes it difficult to estimate the LPL contribution since it depends on the excitation time as well as power (**d**).

Q2. Authors can discuss more about how to obtain suitable energy gap for the design of new LPL materials.

A: We believe that the important factor for designing efficient LPL systems is a small energy gap of $\Delta E(^1\text{CT}-^3\text{LE}_\text{D})$, and many exciplex systems should exhibit LPL if the donor and acceptor pairs satisfy this condition. The $^3\text{LE}_\text{D}$ can be estimated from DFT calculations. The possibility of formation of an exciplex can be inferred from the HOMO level of the donor and the LUMO level of the acceptor. However, estimation of ^1CT is quite difficult because many factors such as molecular structure, orientation, and distance between electron donor and acceptor units can affect the CT state of an exciplex.

At this moment, it is still difficult to discuss efficient molecular design. Future investigation of many exciplex systems will help to create rules for predicting which exciplexes will exhibit efficient LPL emission.

Q3. I noted that the content of donor is 1% in all the work. Whether the different ratios of donor/acceptor influence on the LPL performance?

A: As the reviewer mentioned, the concentration of donor affects the LPL duration. The concentration dependence of TMB/PPT was published in *Nature* **2017**, 550, 384, and we added those of DMDTB/PPT and TTB/PPT in Figure S11. The LPL duration became shorter by increasing the donor concentration. The higher concentration of donor increases the formation of CT state and the probability of charge separation. However, it also accelerates the recombination process due to the higher concentration of donors.

We added a new figure as Figure S11 and the following sentences from page 7 line 16.

“The contribution of delayed fluorescence by triplet-triplet annihilation is almost negligible since the donor concentration is only 1% and the phosphorescence time scale is much shorter than that of LPL. The TTB concentration dependence of the emission spectra and emission decay profiles (Figure S11) are also consistent with exciplex emission. The LPL duration becomes shorter at higher concentrations of donor because the accumulated charges can more easily recombine with donor molecules. For higher donor concentrations, the exciplex emission was slightly redshifted and the room temperature phosphorescence from donors became weaker because of aggregation of donor molecules.”

Figure S11. Donor concentration dependence. Semi-logarithmic plots (a, c) and logarithmic plots (b, d) of the emission decay profiles, steady-state photoluminescence spectra (e, h), and time-resolved spectra (f, g, i, j) of DMTDB/PPT and TTb/PPT with different donor concentrations. Samples were excited for 60 s by a 340-nm LED source with a power of 230 μ W at 300 K. “PL” means the steady-state photoluminescence, “LPL” means the long-persistent luminescence, and “Phos.” means the phosphorescence. The time-resolved spectra were integrated over periods of 1–2 and 10–30 s after stopping excitation.

Q4. More discussion on the comparison between experimental and computational DFT studies can be provided.

A: DFT calculation of donor and acceptor molecules is possible, but calculations of exciplexes is quite difficult since the distance and conformation of both the donor and acceptor affect the energy levels of the exciplex. We have tried to calculate the exciplex energy levels by using the QM/MM method, but we could not obtain reliable results even with the high calculation costs. Thus, we would like to exclude such studies from the paper at this time as we continue to investigate other options.

Q5. To arouse broader interest from the readership in this field, several strong related works can be added, such as donor/acceptor triplet energy transfer (*Adv. Funct. Mater.* 2019, 29, 1807599; *Chem. Sci.* 2017, 8, 590); long-lasting luminescence (*Nat. Commun.* 2018, 9, 2798). These literatures may help readers better understand the development of recent similar luminescent materials.

A: We thank the constructive comment. We added these references (ref. 12, 26 & 27).

Q6. Authors need to discuss the stability of the organic long-persistent luminescent LPL.

A: As the reviewer is aware, stability is important for practical applications. However, since this manuscript focuses on the emission mechanism of OLPL systems, stability issues will be discussed in detail in a future article. For the reviewer's reference, we have found that, although the stability depends on the conditions, the PL and LPL did not degrade after photoexcitation for several hours. Also, the samples were stable under inert atmosphere, as the LPL performance did not change after being stored of 5 months in glovebox while partial exposed to ambient light (Figure R1).

Figure R1. Emission decay profiles of TMB/PPT, DMDTB/PPT, and TTB/PPT after 5-month storage in a glovebox. Excitation condition: 230 μ W, 60 s, 300 K.

Q7. Several mistakes can be avoided, such as page 12, line 242: Wiley Online Library?

A: We are sorry for the mistake. The correction was made accordingly.

Overall, from scientific and technological views, this work can be suitable for Nature Communications after giving revisions based on the comments above.

Answers to the reviewer #2

In this work, the authors suggest the modulation of energy between exciplex state and localized triplet state influences the performance of long-persistent luminescence (LPL). Through the molecular design of the three donors with different energy of HOMO in exciplex system, distinct excited-state pathways show the competition between phosphorescence and exciplex emissions. Though the experimental results seem to be self-consistent with the model, the proposed mechanism is not new. The exciplex systems based on PPT and TMB have been reported in several articles and similar mechanism has been discussed in detail (ref: Adv. Mater. 2018, 30, 1800365; Adv. Mater. 2018, 30, 1803713. and Chem. Lett. 2019, 48, 270-273). Furthermore, organic long persistence luminescence has been intensively studies during the past few years. Numerous papers have been published, which all claimed to have a bright future goal for advanced applications such as display, anti-fake, information, storage and bio-imaging etc. Unfortunately, up to this stage. I do not see any breakthrough in the practical or leaping applications. Note that various other types of inorganic phosphors possess better efficiency of LPL in similar applications proposed. In other words, a solid progress and task of top priority of the relevant research in this field is to show impressive and leading applications, not the explorations of more molecules and perhaps debatable mechanisms but with similar properties. These, together with a number of technical comments listed below, lead me to hesitate recommending this article published in Nature Communication.

A: We would like to disagree with the reviewer's comment that "numerous papers" have been published regarding OLPL. While the reviewer does cite three of our previous papers regarding this area, there are few other works discussing OLPL in the literature at present. On the other hand, we would agree that there are numerous papers about other long-lived luminescence processes, particularly organic room temperature phosphorescence (RTP), and that RTP emitters have yet to provide novel applications.

However, OLPL and RTP are totally different emission mechanisms. OLPL requires an intermediate charge-separated state (or trapped states) as part of the emission process, which leads to a power-law emission decay. By contrast, phosphorescence does not require a charge-separated state and is simply a transition between different spin states, usually from a triplet excited state to the singlet ground state, which follows exponential emission decay. While LPL is long lived because of charge separation and subsequent recombination of initially generated excitons, RTP is long lived because of the low probably of the transition occurring in the initially generated excitons. The extra steps and possibility to generate a variety of excitons upon recombination in the OLPL process makes it significantly more complex than RTP and is the focus of this manuscript.

In the case of TMB/PPT, LPL emission, i.e., emission after recombination of charge separated states, is from ^1CT , so we can interpret this as fluorescence-based LPL emission. On the other hand, the LPL emission of TTB/PPT is a mixture of emission from ^1CT and ^3LE , so we can interpret this as a mixture of fluorescence-based and phosphorescence-based LPL emission. Since these emission processes are different from traditional fluorescence, delayed fluorescence, and phosphorescence, we need to understand the detailed emission mechanism.

Our previous papers regarding LPL emission focused on applications, such as color-tuning

(Adv. Mater. 2018, 30, 1800365) and flexibility (Adv. Mater. 2018, 30, 1803713), and the fabrication process (Chem. Lett. 2019, 48, 270-273), and the detailed emission process is still unclear. This manuscript mainly discusses the detailed emission mechanism of the LPL phenomenon based on the photochemistry. We do not provide new applications in this manuscript but rather provide mechanisms to help understand the charge carrier dynamics. Further understanding of the intermediate CS states is key for the development of novel applications such as CS-based energy storage and photo-rechargeable mechanoluminescence while also aiding in the understanding of the emission mechanism of inorganic LPL materials.

To clarify the difference between phosphorescence and LPL, we added the following sentences and new Figure 1 from page 2 line 3.

“The first LPL emitters were based on inorganic crystals, and performance was greatly improved through doping.¹⁻³ Several charge accumulation mechanisms, such as electron or hole trapping mechanisms, have been proposed to explain inorganic LPL.^{1,3} Unlike phosphorescence, which can also be long lived but is a transition between different spin states (usually from a triplet excited state to the singlet ground state), LPL systems do not follow an exponential decay and usually follow a power-law decay because of the presence of the intermediate states (Figure 1).”

Figure 1. Differences between LPL and phosphorescence. **a.** Schematic diagram of fluorescence, phosphorescence, and LPL. Phosphorescence is a transition from triplet excited state (T_1) to singlet ground state (S_0). LPL is an emission mechanism in which the energy passes through an intermediate states like a trapped state. There is no restriction regarding spin state. While LPL is long lived because of charge separation and subsequent slow recombination (second-order kinetics) of initially generated excitons, phosphorescence is long lived because of the low probability of the transition (first-order kinetics) occurring in the initially generated excitons. **b.** The ideal emission decay profiles of phosphorescence and LPL on logarithmic plots. Phosphorescence follows an exponential decay and LPL a power-law decay.

Q1. The system of TTB/PPT seems to have no solid evidence for the existence of exciplex. If the emission at 435 nm is ascribed to the exciplex fluorescence, then the authors have to explain the first fast decay (0~5 ns) accompanied with the later slow rise (10~30 s) in time resolved emission spectra. To my viewpoint, it seems to be more appropriate that the emission at 435 nm is from singlet LE of donor rendered perhaps by triplet-triplet annihilation (rather slow diffusion in solid)

of triplet LE of donor because emission of TTB/ZEO also exhibits singlet emission at 435 nm. If this is the case, it is irrelevant to the exciplex system.

A: The emission of TTB/PPT is similar to the fluorescence of TTB, but this emission originates from an exciplex between TTB and PPT. The fluorescence lifetime of TTB in ZEONOR is 1.3 ns (Table 1 and Figure S3). In contrast, the prompt emission of TTB/PPT is slightly broader than that of TTB/ZEO and the emission continues for a duration on a microseconds timescale (the reordered Figure S6 (a-c)). The triplet-triplet annihilation (TTA) of TTB is almost negligible since the donor concentration is only 1%. Also, the time scale of delayed fluorescence by TTA is similar to that of phosphorescence since TTA requires triplet excited states. However, the phosphorescence lifetime is 720 ms. Therefore, the slow rise after 10 s originates from the charge recombination processes.

Emission by exciplex fluorescence, exciplex TADF, and TTB phosphorescence from initial excited states without the charge separation/recombination process end after around 5 s, when the initial excited states have been almost completely deactivated. Continuous charge recombination produces both singlet and triplet exciplexes (^1CT and ^3CT). Therefore, the dual emission from both exciplex fluorescence and TTB phosphorescence were obtained after around 5 s. We added the following sentences from page 7 line 16.

“The contribution of delayed fluorescence by triplet-triplet annihilation is almost negligible since the donor concentration is only 1% and the phosphorescence time scale is much shorter than that of LPL. The TTB concentration dependence of the emission spectra and emission decay profiles (Figure S11) are also consistent with exciplex emission. The LPL duration becomes shorter at higher concentrations of donor because the accumulated charges can more easily recombine with donor molecules. For higher donor concentrations, the exciplex emission was slightly redshifted and the room temperature phosphorescence from donors became weaker because of aggregation of donor molecules.”

Q2. The definition of the normalized intensity for transient emission decay curves in Figure S6 and S7 is vague. In Figure S7c, the intensity of DMDTB/PPT and TTB/PPT at 10 μs have dropped to the order of 10^{-4} already. However, the order of the intensity in Figure S7f are still 10^{-2} to 10^{-3} at 10 ms, which is inconsistent and contradicts to the results in Figure S3 to S5. The author should explain this confliction.

A: Thank you for the valuable input. The data in the reordered Figure S7 and S8 (original Figure S6 and S7) are obtained by using a streak camera and normalized by the prompt emission peak intensity. Since the time resolution of the streak camera is around 1/1,000 of the obtained time range, the integrated times for the prompt emission are not same in the different time range. Therefore, we cannot directly compare the intensities among the different time ranges. To clarify these points, we changed the modified Figure S7 and S8 to semi-log plots and add the description in the caption.

Q3. The intensity of the all samples in Figure 1c decays to the order 10^{-2} at 10 to 100 sec. I doubt the normalized intensity ratio can really represent the ratio of emission contributed by LPL

since the intensity ratio is different from the transient emission decay profile (Figure S6 and S7). If the intensity ratio in Figure 1c indeed represents the real ratio of LPL, please explain the different results in Figure S6 and S7. If not, the authors have to elaborate on how much ratio of emission resulted from LPL.

A: As the reviewer mentioned, the ratio of LPL is important information but difficult to calculate. In original Figure 1c (the reordered Figure 2c), each sample was excited by a CW-LED light for 60 s and emission was obtained with a multichannel photodetector with an expose time of 500 ms. This obtained timescale is much longer than that of the streak camera data (nanoseconds to milliseconds, Figures S7 and S8 in the revised manuscript). Moreover, the excitation conditions are different. A pulse laser (20 ps pulse width, 10 Hz) was used for excitation in the streak system. Since the LPL component depends on the excitation time (pulse width) (Fig. S15 and S16), the contribution of the LPL component is different in these measurements. Therefore, we cannot compare these two data directly. To most accurately estimate the contribution of LPL, we need to obtain the emission decay profiles over the nanosecond (10^{-8} s) to tens of hours (10^4) timescales, during which the emission intensity will change over 10 orders of magnitude. The ideal emission decay profile is shown in Fig. S17. However, we could not find a method to obtain such detailed emission decay over such a wide ranges of intensities and time, so this estimate is the best we can presently do.

Figure S15. Excitation power dependence. Semi-logarithmic plots (a, c) and logarithmic plots (b, d) of the emission decay profiles of 1 mol% DMTDB/PPT and 1 mol% TTB/PPT with different excitation powers. Samples were all excited for 60 s (from -60 to 0 s) by a 340-nm LED source at 300 K. “PL” means the steady-state photoluminescence, “LPL” means the long-persistent luminescence, and “Phos.” means the phosphorescence.

Figure S16. Excitation time dependence. Semi-logarithmic plots (a, c) and logarithmic plots (b, d) of the emission decay profiles of 1 mol% DMTB/PPT and 1 mol% TTB/PPT with different excitation times. Samples were all excited by a 340-nm LED source with a power 230 μ W at 300 K. “PL” means the steady-state photoluminescence, “LPL” means the long-persistent luminescence, and “Phos.” means the phosphorescence.

Figure S17. Ideal emission spectra and logarithmic plots of the emission decay profiles of phosphorescent materials (a, b) and OLPL materials (c, d). The phosphorescence quantum yield can be calculated from the areas under the absorption and phosphorescent emission spectra (a). If

the fluorescence and phosphorescence spectra have a large overlap, phosphorescence quantum yield can be estimated from the emission decay profiles because both fluorescence and phosphorescence follow an exponential decay (b). The OLPL system exhibits fluorescence, TADF, phosphorescence, and LPL from similar energy levels (c). Although the fluorescence, TADF, and phosphorescence follow an exponential decay, the power-law decay of LPL makes it difficult to estimate the LPL contribution since it depends on the excitation time as well as power (d).

Q4. The proposed mechanism shown in Figure 3a is dubious. The rate of reverse intersystem crossing (RISC) has been reported to be about 10^5 to 10^6 s⁻¹ in previous literature (Appl. Phys. Lett. 2013, 102, 153306.). From the kinetic point of view, the rate of RISC is much faster than the radiative rate of the LPL (the lifetime is too long). As a result, it is hard to understand why the ³CT state can undergo RISC and emit LPL simultaneously. The RISC should dominate the excited ³CT state behavior. The authors should give a rational explanation.

A: We apologize for the confusion. The LPL is not a simple transition from excited states but the general term for emission in which the energy passed through a charge separation and recombination process.

This emission system can be separated into two parts. The first part is emission from the simple photoexcited states (fluorescence, TADF, and phosphorescence). All processes follow the rate constants of each transition as the reviewer mentioned. The second part is emission from charge recombination (LPL). This charge recombination continually produces excited states after stopping photoexcitation. This charge recombination follows a power-law decay. The charge recombination generates excited states, which then follows the rate constants of each transition. The timescales of the simple transitions and LPL is different. To clarify these processes, we separated the new Figure 4a into two processes.

Figure 4. Proposed emission mechanism before (a) and after (b) recombination of charges and LPL path in TMB/PPT (c), DMDTB/PPT (d), and TTB/PPT (e). The energy levels were calculated from the onsets of the corresponding emission spectra. Abbreviations of electron donor (D), acceptor (A), charge transfer (CT), electron transfer (ET), charge separated state (CSS), charge separation (CS), and charge recombination (CR) are used. The dotted lines represent weaker

luminescence processes and the solid lines stronger ones.

Q5. Continuing the above comment, in Figure 3d, the authors claimed the $^3\text{LE}_D$ state would generate both phosphorescence and LPL. The authors should tell the difference between these two emissions. Moreover, the authors carried out the emission spectra in 10 K showing the stronger phosphorescence from TTB species than that at room temperature (Figure S10). This observation indicates that the triplet exciton would be finally trapped in the $^3\text{LE}_D$ state in the TTB/PPT system. Such statement is very tenuous due to the lack of direct evidence. The non-radiative pathways would be suppressed at low temperature, which would certainly enhance the intensities of the phosphorescence from all triplet states. Therefore, this result cannot support the proposed mechanism in a direct manner. More convincing evidence should be provided.

A: We apologize for the confusion. The new Figure 4 will help to underscore the difference between phosphorescence and LPL. The LPL originates from charge recombination, so created excitons can emit through ^3CT (fluorescence or TADF of exciplex) or $^3\text{LE}_D$ (phosphorescence of donor), just like if the excitons had been generated by hole and electron recombination in electroluminescence. However, because charge recombination is rate-determining process, the kinetics of LPL is totally different from those of normal first-order luminescence processes, e.g., fluorescence and phosphorescence.

Q6. Regarding the DMDTB/PPT system, the authors mentioned that the exciplex emission shows a large spectral shift during the TADF process, which can be attributed to the excited-state conformational change between the two stable isomers of DMDTB. It is believed that the *cis-trans* isomerization is difficult to occur in the solid film. So, what kinds of conformation-dependent emissive properties do the authors expect here? The authors should pay some efforts to make the interpretation clear. Furthermore, the cited reference (ref. 25) is inappropriate because it doesn't provide any related photophysical idea.

A: We are sorry that our explanation was not clear enough. This is not the conventional *cis-trans* isomerization of a double bond since C-N and the center Ph rings are connected by a single bond. Therefore, these two conformers are rotational isomers having very small activation energies. To avoid this misunderstanding, we changed the description of the two isomers and added the activation energy of the isomerization in the revised Figure S14.

As the reviewer mentioned, the original ref. 25 was not appropriate. We think that ref. 18, which investigated the dipole moment of TADF OLEDs, partly supports this concept. A large dipole moment change by excitation induces the dipole orientation of the surrounding molecules from microseconds timescale after removing the electric field. This orientation leads to a redshift and blueshift of the transient emission spectra and the broadening of the steady-state spectra.

Figure S14. The potential energy surface and the conformations of DMDTB at the ground states in vacuum at the B3LPY/6-31G level (refer to the method of [3]). The dipole moments of conformers A and F were calculated using DFT at the PBE1PBE/ma-Def2-TZVP level.

Minor

Q1. Please change the “Figure 2” in page 4 line 80 to Figure 1. “Figure 2 shows the steady-state photoluminescence and time-resolved (1–2 s, 4–5 s, 10–30 s, and 100–300 s after stopping excitation)”

A: We are sorry for the mistake. A correction was made accordingly.

Answers to the reviewer #3

This manuscript from Adachi and coworkers follows up on this groups very interesting and significant report in 2017 that long-lived luminescence can be obtained from organic exciplexes. The present manuscript examines three combinations of electron donors with an electron acceptor in order to understand how the electronic properties of the donor affect OLPL performance. This is an important study as the results can be used to guide the design of future exciplex systems. The authors reach the conclusion that the energy gap between the lowest localized triplet and lowest charge transfer singlet is an important design criterion, with a larger gap to reduced efficiency. It is not completely clear to this reviewer how this conclusion is supported by the data:

In Table 1 the PL quantum efficiencies of the three donor/acceptor blends are given along with the energy gap. The three blends show rather similar quantum efficiencies, but this is apparently the overall quantum efficiency for PL, not just OLPL. What is the OLPL quantum yield in each of these blends? This is the key parameter assessing performance of the OLPL material. The data shown in Figure 1c suggest that there is a difference in OLPL efficiency, but this is not quantitated, and it is not clear whether these decays are normalized or not. This is a key issue that the authors must address prior to publication.

A: We thank the referee for taking the time to read and evaluate our manuscript. Although the photoluminescence quantum yield (PLQY) is an important factor when discussing conventional photoluminescence, defining a similar value, e.g., an LPL quantum yield, is difficult because of the charge accumulation process. This is why quantum efficiencies are not reported for inorganic LPL systems even in academic research.

In the case of the conventional photo emitters, the excited states are generated only by the initial photoirradiation. All emission processes are first-order reactions that follow exponential decay. Therefore, PLQY can be calculated from the area under the absorption and area under the emission spectra obtained by using an integration sphere. (Figure S17).

Phosphorescence quantum yield can also be calculated from the area under the absorption spectrum and the area under the steady-state emission spectrum, this time only considering phosphorescent emission (Figure S17a). Even if the fluorescent and phosphorescent spectra overlap, we can calculate the ratio of fluorescence and phosphorescence from the emission decay profiles since all processes follow exponential decay (Figure S17b).

In contrast, LPL systems store excited energy through a charge accumulation process and generate new excited states from the accumulated charges over a long period after turning off the photoexcitation. This process is a second-order reaction, so it follows a power-law decay. Since the charge separation and recombination processes are very slow, LPL emission is not fully reflected in the steady-state emission spectra (Figure S17c).

Furthermore, OLPL systems exhibit fluorescence, TADF, phosphorescence, and LPL from similar energy levels. Since the emission decay profile of the OLPL system is very complicated, even in the ideal case, observation of the emission decay profile from nanosecond (10^{-8} s) to tens of hours (10^4 s) timescales are required to calculate the LPL contribution (Figure S17d). On top of that, the LPL component depends on the excitation conditions (time and intensity) (Fig. S15 and S16). Therefore, we cannot provide an LPL quantum efficiency.

We added the following sentences from page 9 line 16.

“Notably, the presented photoluminescence quantum yields (Φ_{PL}) do not completely reflect the LPL components (Table 1). The quantum efficiency of LPL emission is difficult to define because the charge accumulation and release processes are slow and complicated in contrast to those of long-lived phosphorescence. Furthermore, the LPL emission depends on the excitation time as well as the excitation power, while the phosphorescence component is constant (Figure S15 and S16). Because the LPL system continuously provides the new excited states after turning off the photoexcitation, we cannot calculate the Φ_{PL} from the steady-state photoluminescence spectra (Figure S17). This is why Φ_{PL} is not discussed even in inorganic LPL materials.^{1,2,3}”

Figure S17. Ideal emission spectra and logarithmic plots of the emission decay profiles of phosphorescent materials (**a, b**) and OLPL materials (**c, d**). The phosphorescence quantum yield can be calculated from the areas under the absorption and phosphorescent emission spectra (**a**). If the fluorescence and phosphorescence spectra have a large overlap, phosphorescence quantum yield can be estimated from the emission decay profiles because both fluorescence and phosphorescence follow an exponential decay (**b**). The OLPL system exhibits fluorescence, TADF, phosphorescence, and LPL from similar energy levels (**c**). Although the fluorescence, TADF, and phosphorescence follow an exponential decay, the power-law decay of LPL makes it difficult to estimate the LPL contribution since it depends on the excitation time as well as power (**d**).

To clarify the difference between phosphorescence and LPL, we added the following sentences and new Figure 1 from page 2 line 3.

“The first LPL emitters were based on inorganic crystals, and performance was greatly improved through doping.¹⁻³ Several charge accumulation mechanisms, such as electron or hole trapping mechanisms, have been proposed to explain inorganic LPL.^{1,3} Unlike phosphorescence, which can also be long lived but is a transition between different spin states (usually from a triplet

excited state to the singlet ground state), LPL systems do not follow an exponential decay and usually follow a power-law decay because of the presence of the intermediate charge accumulated states (Figure 1).”

Figure 1. Differences between LPL and phosphorescence. **a.** Schematic diagram of fluorescence, phosphorescence, and LPL. Phosphorescence is a transition from triplet excited state (T_1) to singlet ground state (S_0). LPL is an emission mechanism in which the energy passes through an intermediate states like a trapped state. There is no restriction regarding spin state. **b.** The ideal emission decay profiles of phosphorescence and LPL on logarithmic plots. Phosphorescence follows an exponential decay and LPL a power-law decay.

Other points for the authors to address:

Q1: It would be useful to provide a comparison of the efficiency of these systems relative to the conventional inorganic/polymer systems discussed in the introduction.

A: We thank the reviewer for this comment. We added a comparison between a commercial inorganic LPL product and the current best OLPL system (ref. 7) (**Figure S1**). The inorganic LPL material exhibits a small drop of the emission intensity after turning of the photoexcitation since the emission is mainly originates from the charge separation and recombination processes. In contrast, OLPL system exhibits a large drop because most of the initial emission originates from simple photoexcited states that do not pass through charge separation and recombination. We added the relative sentence in page 2 line 21 as follows.

“However, a large performance gap still exists between the present OLPL system and the commercial high-performance inorganic LPL products (Figure S1).”

Figure S1. Comparison between OLPL and inorganic LPL system. Semi-logarithmic plots (a) and logarithmic plots (b) of the emission decay profiles of the reported OLPL system (1 mol% m-MTDATA/PPT)^[1] and a commercial inorganic LPL material (Super α -Flash, LTI corporation, Japan). Emission spectra during photoexcitation (c) and after the excitation (d) of inorganic LPL material. All samples were 1 cm² and were excited for 60 s by a 340-nm LED source with same power 230 μ W at 300 K.

Q2: Figure 3 – why are some decays dotted lines and some solid lines? There doesn't seem to be any logic to this and no legend is provided. The Figure caption is also unclear – each frame should be clearly labelled in the caption.

A: We apologize about this unclear figure. The dotted lines represent luminescence processes that are weaker than those represented by the solid lines. We have added this information in the caption.

Q3: Lines 178-180 – there are examples of pure organic systems that show RTP – the statement regarding efficiency of these should be referenced.

A: While the efficiency of the phosphorescence can influence the overall emission, the point that we really wanted to highlight here is that excitons in the ³LE state are less likely to undergo charge separation, so excitons that get trapped on a low ³LE_D are unlikely to separate and contribute to LPL. Thus, a large energy gap of $\Delta E(^1\text{CT}-^3\text{LE}_D)$ will weaken the LPL while a smaller energy gap will allow ³LE_D excitons to convert to ¹CT, increasing the chance for separation and subsequent emission as LPL. We changed the description in the main text (page 9 line 9) as follows.

“These results clearly indicate the importance of the energy level of ³LE_D for obtaining

efficient LPL emission. Since 3LE excitons are less likely to undergo the charge transfer step needed for creating separated charges that contribute to LPL, the higher exciton population on 3LE_D induced by a large energy gap of $\Delta E(^1CT-^3LE_D)$ will reduce the number of excitons that can convert into CS states. Thus, efficient LPL emission requires a small energy gap to ensure a higher number of 1CT excitons that can contribute to the accumulation of separated charges.”

Q4: Line 80 – Figure 1 not Figure 2.

A: We are sorry for the mistake. This was corrected accordingly.

Reviewers' comments:

Reviewer #1 (Remarks to the Author):

In the revised manuscript, Adachi and co-workers have carefully answered all the questions from reviewers, and I am pleased to recommend to accept this work as it is.

Reviewer #2 (Remarks to the Author):

I have carefully examined the replies and corresponding changes. Although the authors made certain efforts to revise the manuscript, to my viewpoint, the experimental results still cannot directly support the mechanism of LPL and the interpretations of the mechanism are still rather confusing. The authors claim that the charge-separated state (CSS) plays a significant role in LPL, which is able to discriminate LPL from phosphorescence. However, there isn't any direct evidence to prove that the long-lived CSS exists exactly since both polynomial functions and exponential functions can fit the experimental data. It is unreliable to conclude the existence of CSS by only the time-resolved emission profiles without any other direct supports.

In addition, all the emission decay profiles neglect the spectral shift of time-resolved emission spectra at the specific wavelength (Figure 3C), so that the LPL resulting from phosphorescence is still not convincing. Furthermore, in Figure 4b, the authors state that the LPL results from the transition of CT1- S0 and LE3- S0, indicating that the luminescence after charge separation still consists of phosphorescence. I have no idea how to differentiate the phosphorescence before and after charge separation based on the current experimental results. The authors should provide physical meanings about the kinetics behind the CSS states, which can be fitted by the power law model.

Last but not the least, the novelty of this work comparing to the three references I mentioned previously is my major concern. In the reference *Adv. Mater.* 2018, 30, 1803713, a similar mechanism has been proposed in its Figure 3, showing that LPL is rendered by the emission from CT1 (the S1 state of exciplex) and T1 in TMB corresponding to the CT1 and LE3 in this work, respectively. There is no new or more clear insight into the mechanism. Together with numbers of physical interpretations that are debatable, I cannot recommend this manuscript for publication in *Nature Communications*.

Reviewer #3 (Remarks to the Author):

Reviewer #3 provided notes to the editor, in which he/she recommended publication.

I have carefully examined the replies and corresponding changes. Although the authors made certain efforts to revise the manuscript, to my viewpoint, the experimental results still cannot directly support the mechanism of LPL and the interpretations of the mechanism are still rather confusing. The authors claim that the charge-separated state (CSS) plays a significant role in LPL, which is able to discriminate LPL from phosphorescence.

However, there isn't any direct evidence to prove that the long-lived CSS exists exactly since both polynomial functions and exponential functions can fit the experimental data. It is unreliable to conclude the existence of CSS by only the time-resolved emission profiles without any other direct supports.

As we have proved the presence of CSS using transient absorption measurements in our previous publications, which are referenced throughout the manuscript, we did not include a detailed discussion of their presence here. However, in light of the reviewer's comment, we have added time-resolved ESR data to further support the presence of CSS. As shown in new Fig. S3, the ESR signal clearly increases after photo-excitation. The ESR signal would only increase like this if there are radicals in the film, indicating that charge separation is taking place to form radical states.

Figure S3. ESR signal of TMB/PPT system. The ESR signal clearly increases after photo-excitation due to the charge separation process. The ESR signal is gradually decrease by time.

In addition, all the emission decay profiles neglect the spectral shift of time-resolved emission spectra at the specific wavelength (Figure 3C), so that the LPL resulting from phosphorescence is still not convincing.

The presented emission decay profiles are not the intensity at a single wavelength but in fact the integrated emission intensity over most, if not all, of the emission spectra. The emission decay profiles in Figs. 1, S11, S15, and S16 contain all of the emission over wavelengths from 400 to 900 nm. The emission decay profiles obtained with a streak camera system (Figs. S7 and S8) were integrated over the emission from 400 to 650 nm. We added this information into figure caption.

To clearly separate the phosphorescence originating from initially generated excited states from that originating from CSS, we measured decay profiles of TTB/PPT for only the main peak of phosphorescence. This film was chosen because the phosphorescence and fluorescence components have the least overlap. As can be seen in Fig. R1, the phosphorescence obtained at 550 ± 5 nm exhibits a clear change in decay as it switches from the initial component to the LPL component (originating from CSS). Thus, we can state that this is phosphorescence via CSS (i.e., LPL) and not a typical long-lived phosphorescence.

Figure R1. Emission decay profile of a TTB/PPT film obtained at 550 ± 5 nm in semi-log (a) and log-log plots (b).

Furthermore, in Figure 4b, the authors state that the LPL results from the transition of CT1- S0 and LE3- S0, indicating that the luminescence after charge separation still consists of phosphorescence. I have no idea how to differentiate the phosphorescence before and after charge separation based on the current experimental results. The authors should provide physical meanings about the kinetics behind the CSS states, which can be fitted by the power law model.

Although the phosphorescence spectra before and after CS are the same, they have different emission decay profiles that span different timescales, as shown in Fig. S19. Without the CS process (conventional phosphorescence), the transition is controlled by the rate constant of the transition from T_1 to S_0 . By contrast, with CS, since the charge recombination is much slower than that of the transition from T_1 to S_0 , the emission is dominated by the CS process, which follows power-law kinetics.

The power-law kinetic results (power-law kinetic, $I(t) \propto t^{-m}$, $m = 0.1-2$) from charge recombination can be explained by several physical models discussed in previous literatures about LPL from organic molecules (TMB/poly(alkyl methacrylate)s)¹ and thermoluminescence of the inorganic LiF^2 and the organic molecule polyethylene terephthalate³. These models can be separated into the diffusion model and the electron tunneling model of geminate ion recombination. The following is a brief introduction about several models.

In the diffusion model, we consider the distribution of electrons (radical anions) after the charge separation process. The Debye-Edward model⁴ can explain $I(t) \propto t^{-m}$, $m = 1$, but cannot explain $m > 1$. Abell and Mozumder⁵ gave a revised model showing that, when $t \rightarrow \infty$, m is 1.5, which can explain the change of m value over time. The Hong-Noolandi model⁶ can also explain the recombination rate $R(t) \propto t^{-m}$ with $m = 1.5$. Stolzenburg, Ries and Bässler⁷ proposed $I(t) \propto t^{-m}$, $m = 1$, based on the Hong-Noolandi model by considering the energetic relaxation of carriers subject to random walk. Although the m value is changed according to the model, all models explain the power-law decay.

The electron tunneling model is mainly used to explain the isothermal recombination luminescence at low temperatures for irradiated organic compounds. For example, the Tachiya-Mozumder model shows $I(t) \propto t^{-m}$ with m very close to unity over a wide time range.⁸

TMB/poly(alkyl methacrylate)s irradiated by a laser at low temperature exhibited LPL containing fluorescence and phosphorescence at same time (Figure 4 in reference 1).¹ The emission decay profiles of both fluorescence and phosphorescence follow a power-law decay ($I(t) \propto t^{-m}$, $m \approx 1$) due to the presence of CSS. In this system, there is no CT state, but charge separation is realized by the second-photon absorption from the excited states (Fig. R2a). The charge recombination directly generates both singlets and triplets on TMB and exhibits both fluorescence and phosphorescence following power-law decay.

Our OLPL system also exhibits emission decay kinetic ($I(t) \propto t^{-m}$, $m \approx 1$) which can be explained by aforementioned models. In contrast to the previous results, the presence of electron donor and acceptor helps the charge separation and generates geminate charge pairs under weak photoexcitation (Fig. R2b).

Figure R2. (a) LPL by two-photon absorption (b) LPL by charge separation.

1. H. Ohkita, W. Sakai, A. Tsuchida, M. Yamamoto, *Macromolecules* **1997**, *30*, 5376-5383.
2. C. A. Boyd, *J. Chem. Phys.* **1949**, *17*, 1221-1226.
3. Y. Hama, Y. Kimura, M. Tsumura, N. Omi, *Chem. Phys.* **1980**, *53*, 115-122.
4. P. Debye, J. O. Edwards, *J. Chem. Phys.* **1952**, *20*, 236-239.
5. G. C. Abell, A. Mozumder, *J. Chem. Phys.* **1972**, *56*, 4079-4085.
6. K. M. Hong, J. Noolandi, *J. Chem. Phys.* **1978**, *68*, 5163-5171.
7. F. Stolzenburg, B. Ries, H. Bässler, *Ber. Bunsenges. Phys. Chem.* **1987**, *91*, 853-858.
8. M. Tachiya, A. Mozumder, *Chem. Phys. Lett.* **1975**, *34*, 77-79.

We added the following sentences from page 3 line 9.

“The power-law kinetic results (power-law kinetic, $I(t) \propto t^{-m}$, $m = 0.1-2$) from charge recombination can be explained by several physical models discussed in previous literatures about LPL from organic molecules (TMB/poly(alkyl methacrylate)s)¹³ and thermoluminescence of the inorganic LiF¹⁴ and the organic molecule polyethylene terephthalate¹⁵. These models can be separated into the diffusion model^{9,10,16,17} and the electron tunneling model¹⁸ of geminate ion recombination. In the diffusion model, we consider the distribution of electrons (radical anions) after the charge separation process. The electron tunneling model is mainly used to explain the isothermal recombination luminescence at low temperatures for irradiated organic compounds.”

Last but not the least, the novelty of this work comparing to the three references I mentioned previously is my major concern. In the reference *Adv. Mater.* 2018, 30, 1803713, a similar mechanism has been proposed in its Figure 3, showing that LPL is rendered by the emission from CT1 (the S1 state of exciplex) and T1 in TMB corresponding to the CT1 and LE3 in this work, respectively. There is no new or more clear insight into the mechanism. Together with numbers of physical interpretations that are debatable, I cannot recommend this manuscript for publication in *Nature Communications*.

As the reviewer mentioned, we proposed that the contribution of ³LE (phosphorescence) in LPL in the previous publication (*Adv. Mater.* 2018, 30, 1803713). However, we did not prove the effect of $\Delta E(^1CT-^3LE_D)$ on LPL emission since the focus of the previous paper is the demonstration of flexibility and processability based on polymers. Therefore, we clearly indicated this fact in the introduction (page 3, line 13) and aimed to analyze the complicated LPL phenomenon. Since LPL systems consist of an electron donor and an acceptor, we need to consider the energy levels of ¹LE_D, ³LE_D, ¹CT, ³CT, ¹LE_A, and ³LE_A. In this manuscript, we experimentally demonstrated the contributions of all these levels for the LPL emission. We also proved the presence of TADF in the LPL system. Since OLPL is a new phenomenon and the emission process has still not been explained in detailed, we believe this manuscript will help to understand the unusual long-lived charge separated state.

REVIEWERS' COMMENTS:

Reviewer #2 (Remarks to the Author):

I have carefully read the letter, replies and corresponding changes. This time the authors provide detailed information to verify the existence of LPL and the difference between LPL and RTP. I am satisfied with these replies and changes. This article is recommended to publish in Nat. comm.